# Dermal Pericytes Exhibit Declined Ability to Promote Human Skin Regeneration with Ageing in 3D Organotypic Culture Models

**DOI:** 10.3390/cells10113051

**Published:** 2021-11-06

**Authors:** Lizhe Zhuang, Rahul M. Visalakshan, Pritinder Kaur

**Affiliations:** 1Epithelial Stem Cell Biology Laboratory, Research Division, Peter MacCallum Cancer Centre, Melbourne, VIC 3002, Australia; LZ377@MRC-CU.cam.ac.uk; 2Medical Research Council Cancer Unit, Hutchison/MRC Centre, University of Cambridge, Cambridge CB2 0XZ, UK; 3Epithelial Stem Cell Biology Laboratory, Curtin Medical School, Curtin Health Innovation Research Institute, Curtin University, Perth, WA 6102, Australia; rahul.visalakshan@curtin.edu.au

**Keywords:** skin, dermis, pericyte, microenvironment, human, ageing

## Abstract

The well documented decline in the regenerative ability of ageing human skin has been attributed to many factors including genomic instability, telomere shortening, poor nutrient sensing, cellular senescence, and stem cell exhaustion. However, a role for the dermal cellular and molecular microenvironment in skin ageing is just emerging. We previously showed that dermal pericytes co-operate with fibroblasts to improve human skin regeneration in an organotypic skin culture model, and even do so in the absence of fibroblasts. Here, we report that the number of dermal cells, particularly pericytes, declines significantly in human skin of donors aged > 50 years. Notably, aged pericytes promoted epidermal regeneration of neonatal keratinocytes in organotypic cultures and the resulting epithelium exhibited a Ki67^+^/ΔNp63^+^ basal layer and terminal differentiation. However, the epithelium lacked several features of homeostasis displaying lower levels of ΔNp63 expression, decreased LAMA5 deposition at the dermo-epidermal junction, and the absence of basement membrane and hemi-desmosome assembly. We conclude that a decline in pericyte incidence and function contribute to an impaired epidermal microenvironment and poor skin regeneration with ageing in the human skin.

## 1. Introduction

It is well accepted that the dermis acts as a niche for the interfollicular and hair follicle epidermis of the skin, mediated by an array of extracellular matrix (ECM) proteins and growth factors [1]. The epidermal niche also extends to neighbouring epidermal cells in the proliferative basal and suprabasal layers into which they migrate during differentiation. Thus, changes in mechanical stress and cell density also act as signals for cell replacement in development and homeostasis [2,3]. Key regulatory pathways that govern epidermal regeneration have been identified using elegant, genetically engineered mouse models, and 3D organotypic cultures (OCs), reinforcing the idea that microenvironmental signals including the wnt, Notch, Shh, insulin, TGFB, IL-1, and KGF/FGF-7 pathways are critical in tissue maintenance and repair [4,5,6,7,8,9,10,11,12,13] reviewed in [14].

At a cellular level, the role of dermal fibroblasts in promoting epidermal cell proliferation and differentiation ex-vivo in both 2D [15] and 3D OC models [16] was clearly demonstrated by the absolute dependence of epidermal cells on dermally secreted factors, including ECM proteins, e.g., collagens, laminins, and proteoglycans, and paracrine growth factors. Attempts to identify functionally relevant fibroblast subsets that promote epidermal regeneration during development, wound repair, and homeostasis have been undertaken based on physical location, lineage mapping, and RNA seq analysis [17,18,19,20,21,22,23,24]. In mice, a subset of perivascular fibroblasts has been identified that contribute to scarring by excess collagen production [25]. Our studies in neonatal human skin revealed a new class of dermal cells, i.e., pericytes, known to have mesenchymal stem-cell like properties, that promote epidermal regeneration in 3D OCs [26] more potently than dermal fibroblasts. Specifically, pericyte populated OCs exhibited a high epidermal proliferative index attributed to an increase in planar cell divisions mediated by BMP-2 secretion, retaining more cells in the proliferative compartment; and increased LAMA5 deposition evident in basement membrane/hemi-desmosome assembly at the epidermal–dermal junction of OCs, consistent with the demonstration that pericytes secrete LAMA5 [26]—features not observed in fibroblast co-cultures [27]. Thus, we demonstrated a novel function for dermal pericytes as microenvironmental regulators of skin regeneration in addition to their classic role in regulating blood vessel structural stability and permeability [28,29]. Our findings are consistent with the emerging role for pericytes in regenerative medicine, i.e., their widespread distribution via the vasculature enabling contribution to tissue repair through transdifferentiation into other mesenchymal cell types and secretion of paracrine mediators [30]. Our previous work has focused on neonatal skin pericytes; thus here, we sought to determine if skin pericytes exhibited altered functional characteristics with ageing, specifically the ability to promote skin regeneration.

## 2. Materials and Methods

### 2.1. Human Tissue Collection

Human neonatal (1–4 weeks old) foreskin tissue was collected following routine elective circumcision with informed parental consent. Adult human mammary skin was collected from female donors (17–61 years old) without cancer history, undergoing elective plastic surgery for breast reduction with patient consent and ethics approval.

### 2.2. Isolation and Culture of Keratinocytes, Dermal Pericytes, and Fibroblasts

Neonatal foreskin and adult female breast skin tissue were digested enzymatically to obtain epithelial sheets which were trypsinised with vigorous pipetting for 5 min to obtain a basal keratinocyte cell suspension, as described previously [31]. Primary keratinocytes without prior culture were used in all experiments. The corresponding dermal tissue pieces were minced further and incubated in dispase and collagenase at 37 °C for 90 min (neonatal foreskin) or 2 h (adult breast skin) to obtain dermal cell suspensions; and the pericyte and fibroblast fractions obtained by FACS; 1 × 10^7^ dermal cells/mL incubated with antibodies to VLA-1 (1:20, Serotec, Oxford, UK; MCA1133F) and CD45 (1:40, Beckton Dickinson, East Rutherford, NJ, USA; D555483) for an hour and propidium iodide (1:200, Life Technologies, Carlsbad, CA, USA; P3566) to permit exclusion of dead cells. CD45^−^ cells were fractionated into VLA-1^bri^ dermal pericytes and VLA-1^dim^ fibroblasts as validated previously [26,27] and the dermal subsets were reanalyzed to verify purity. Pericytes and fibroblasts were cultured in EGM-2 (Lonza, Basel, Switzerland; CC-3162) and DMEM containing 10% fetal bovine serum (DMEM-10), respectively, at 37 °C with 5% CO_2_.

### 2.3. Organotypic Cultures

OCs were conducted as described [27]. Bovine collagen type I was a kind gift from Organogenesis Inc., Boston, MA, USA. Briefly, cultured neonatal or adult pericytes from skin donors aged ~50 years or over or neonatal fibroblasts between p4 and p6 were harvested at ~90% confluence to prepare DEs. For this, 7.5 × 10^4^ pericytes or fibroblasts were used per DE; or both fibroblasts and pericytes at the ratios indicated for each experiment; 5 × 10^4^ primary uncultured human neonatal keratinocytes, resuspended in 30 μL of EpiLife (Life technologies, M-EPI-500-CA and S-001-5) were seeded on top of each DE. OCs were maintained in Epidermalization Medium for a week and then at an air–liquid interface in Cornification Medium for a week and then Maintenance Medium for another week prior to harvest [31].

### 2.4. Immunostaining

OCs and skin tissue (~12 × 3 mm pieces) were fixed in 10% neutral buffered formalin (45 min at room temperature/RT), and paraffin embedded to obtain 4-μm sections. Sections were de-waxed and hydrated prior to performing antigen retrieval in a pressure cooker at 125 °C for 3 min in Tris-EDTA buffer, pH 9.0. Sections were stained for Ki67 (1:200, Dako, Carpinteria, CA, USA; M7240), ΔN-p63 (1:400, BioLegend, San Diego, CA, USA; 619001), PDGFRβ (1:50, Abcam, Cambridge, UK; ab32570) and collagen IV (1:400, Abcam, ab6586, a kind gift from Dr Mark Fear, University of Western Australia, Crawley, Austrilia). Immunohistochemistry or immunofluorescence was performed using ImmPRESS™ kits (Vector Laboratories, Burlingame, CA, USA), Rabbit Specific HRP/AEC (ABC) Detection IHC Kit (ab93705) for collagen IV or 1 μg/mL DAPI and fluorophore-conjugated antibodies (Life Technologies, A-21241, A-21134, A-21428, A-11006 or A-21124) at 1:200 for 1 h at RT. For LAMA5 detection, paraformaldehyde fixed tissue was processed to obtain 10-μm cryosections incubated with 4C7 monoclonal hybridoma supernatant and secondary fluorescent antibody.

### 2.5. Transmission Electron Microscopy

Subsequently, 3 × 6 mm pieces of OCs from two independent experiments were fixed in 2.5% glutaraldehyde, 2% paraformaldehyde in 0.1 M cacodylate buffer for 2 h at RT. They were then washed in cacodylate buffer (3 × 10 min), post-fixed in 2% OsO4 and the washes repeated. After rinsing in distilled water (3 × 10 min), OC pieces were dehydrated in ethanol, treated with acetone (2 × 30 min), and 1:1 Acetone/Spurr’s resin (2 × 2 h) prior to impregnation with 100% Spurr’s resin (2 × 2 h) under vacuum and embedded for ultrathin sectioning. Transmission electron microscopy was performed using a JEOL 1011 (JEOL USA, Inc., Tokyo, Japan).

### 2.6. Quantification and Statistics

Pericyte numbers in breast skin from donors aged 17–50 years were quantitated by immunofluorescent staining of paraffin-embedded sections for the pericyte marker PDGFRβ and DAPI. The upper part of the dermis defined as the area within a depth of 0.2 mm from the epidermal–dermal junction was selected as the region of interest and the area determined by ImageJ. The total number of cells (DAPI^+^) and PDGFRβ^+^ pericytes within the region of interest was counted and normalized against the area as number of cells per mm^2^ dermis. Data was obtained from 9 random fields derived from 3 individual donors within specific age groups, i.e., 17–30 years, 31–40 years, and >50 years. Epidermal thickness of OCs and staining for Ki67 and ΔN-p63 was quantitated per length (µm) of the epithelium or stratum basale in OCs using ImageJ. Quantitative results from immunostaining and OC experiments were replicated in at least 3 independent experiments, and representative samples displayed when reproducible results were obtained.

Statistical analysis was performed using unpaired *t*-test; *p* values < 0.05 were considered statistically significant. Error bars represent standard deviation (SD). All statistical analyses were performed using the software Prism version 6.0.

## 3. Results

### 3.1. Aged Human Skin Exhibits a Decline in Pericyte Incidence

Initially, we assessed dermal cellularity and the incidence of pericytes in human breast skin from older adult donors (~50 years old) compared to neonatal human foreskin. Flow cytometric analysis of freshly disaggregated dermal cells isolated from neonatal (*n* = 13) and older adult skin donors aged ~50 years or over (*n* = 8; donors aged 48, 50, 54, 54, 54, 58, 59, and 60 years old [yo]; mean = 54.6 yo) was performed after immunostaining for CD45 and CD49a to identify the CD45^−^CD49a^dim^ dermal fibroblast population and the CD45^−^CD49a^bri^ pericyte population, previously validated by us [26,27]. Analysis of both single representative samples of adult versus neonatal skin samples (Figure 1A), multiple samples from each age group (Appendix A), and pooled samples (Figure 1B,C; Appendix A) revealed that: (i) CD45^−^ dermal cells were present in higher numbers in neonatal foreskin (~90%) compared to adult breast skin (~40%), concomitant with an increase in CD45^+^ cells in adult skin (~55%) compared to neonatal skin (~8%; Figure 1B; *p* < 0.001); whereas (ii) the CD45^−^CD49a^bri^ pericyte fraction was significantly lower in older adult breast skin (4.2 ± 1.1%) compared to neonatal foreskin (21.5 ± 1.6%) expressed as a percentage of CD45^−^ dermal cells (Figure 1B; Appendix A; Figure 1C, right panel; *p* < 0.0001); and (iii) the CD45^−^CD49a^dim^ fibroblast fraction was similar in both (Figure 1A, right panel; Figure 1C, left panel). Since this difference in pericyte numbers could potentially be attributed to differences in anatomical site (foreskin versus breast), adult breast skin samples from donors aged 17–61 yo were analysed by in situ immunostaining for the well accepted pericyte marker PDGFRβ (Figure 1D), quantitating the numbers of PDGFβR^+^ cells using Image J, and restricting our analysis to a depth of 0.2 mm from the dermo-epidermal junction to ensure minimal variation. A significant decrease in pericyte numbers was observed in skin samples from donors aged over 50 years compared to skin from 17–30 yo and 31–40 yo donors (Figure 1E, left panel; 9 random fields from each of 3 donors per age group, *p* < 0.001), when expressed as the number of PDGFRβ^+^ pericytes/dermal area. Interestingly, enumerating the %PDGFRβ^+^ pericytes as a function of CD45^−^ dermal cells revealed a statistically significant decrease in dermal pericyte incidence as early as 31–40 yo, with a further decrease in donors aged over 50 years (Figure 1E, right panel, *p* < 0.05).

### 3.2. The Functional Capacity of Aged Pericytes to Promote Skin-Regeneration in Co-Operation with Neonatal Fibroblasts Is Affected by Their Lower Incidence

Previously, we reported that neonatal pericytes can significantly enhance the skin regenerative properties of neonatal keratinocytes that have exited the stem/progenitor compartment of the basal layer and committed to differentiation, over and above that observed with neonatal fibroblasts alone in OCs [26]. In order to test whether aged pericytes had similar or diminished epidermal regenerative properties, we established OCs with dermal equivalents populated with aged pericytes (donors aged > 50 yo), seeded with neonatal keratinocytes. The underlying strategy was to keep all other cellular elements besides pericytes of neonatal origin i.e., highly regenerative, while testing pericytes from ageing donors. Given the decrease in pericyte numbers in aged skin, we seeded OCs with either (i) 2% pericytes reflecting their incidence in aged skin; or (ii) 20%, the average pericyte incidence in neonatal skin, to ask if their functional capabilities were retained and not simply a reflection of their dwindling numbers. Figure 2A–C shows the epidermis regenerated from neonatal keratinocytes in a dermal microenvironment of neonatal fibroblasts and either 2 or 20% 54 yo freshly isolated uncultured pericytes in OCs. Histological and quantitative evaluation of the regenerated epidermis indicated virtually no impact of including 2% 54 yo primary pericytes (Figure 2A vs. Figure 2B), whereas increasing the number of pericytes to 20% pericytes (Figure 2C), yielded a significant increase in epidermal thickness compared to fibroblasts alone (Figure 2D). This increased epidermal regenerative ability obtained by the artificial inclusion of greater numbers (20%) of primary pericytes was also observed with a second skin 55 yo donor (Figure 2E vs. Figure 2F,G). In subsequent experiments, we observed that cultured pericytes (p4) from a 54 yo donor also retained the ability to promote epidermal regeneration from neonatal keratinocytes (Figure 2H–J). These data were replicated with 2% versus 20% cultured pericytes (p5) from a second skin donor aged 59 yo (data not shown), suggesting that the decline in pericyte numbers impacts their ability to restore skin regeneration as a function of age compared to fibroblasts alone.

### 3.3. Pericytes from Ageing Human Skin Retain the Ability to Promote Epidermal Regeneration as the Sole Mesenchymal Element in OCs

We recently reported that neonatal pericytes not only promote neonatal epidermal regeneration as the sole element of the dermal equivalents in OCs, but also confer features of homeostatic proliferation, tissue renewal, and differentiation [27], not observed in dermal fibroblast co-cultures. We therefore compared the effects of pericytes from donors aged ~50 yo versus neonatal pericytes, to ascertain any intrinsic functional changes in their ability to regulate epidermal regeneration derived from neonatal keratinocytes, including neonatal fibroblasts as a control. OCs populated with aged pericytes from 3 different donors (48, 54, and 58 yo) and neonatal keratinocytes revealed that despite ageing, pericytes retained the ability to promote epidermal regeneration as judged by histological analysis, i.e., a multilayered epithelium (Figure 3A) with a distinct, polarized basal layer (Figure 3B), giving rise to a multilayered, stratified epidermis exhibiting terminal differentiation, comparable to that induced by neonatal pericytes, but exceeding the inductive properties of neonatal fibroblasts, the latter reported by us previously [27]. Immunostaining for epidermal cell proliferation markers Ki67 and ΔNp63 (Figure 3C,D) showed no difference in the number of Ki67^+^ or ΔNp63^+^ cells/mm within the basal layer of the epidermis in OCs established with neonatal versus aged pericytes (Figure 3E,F), although both Ki67^+^ or ΔNp63^+^ cell numbers were statistically significantly higher than those observed with neonatal fibroblasts, as expected [27]. However, the levels of ΔNp63/cell were much lower in all OCs populated with aged pericytes compared to those populated with neonatal pericytes (Figure 3D). Consequently, we investigated whether other indicators of epidermal homeostasis were perturbed in the OCs containing aged pericytes.

### 3.4. Aged Pericytes Do Not Restore Epidermal Cell Homeostasis

We have reported that recombinant LAMA5 restores the skin regenerative ability of differentiated neonatal human keratinocytes co-cultured with fibroblasts in OCs [32]. Moreover, LAMA5 is secreted by both basal keratinocytes and dermal pericytes, but not fibroblasts in native neonatal human skin [26]. Our recent work suggests that the amount of LAMA5 synthesised by neonatal keratinocytes is rate-limiting for hemi-desmosome/basement membrane assembly and skin regeneration in OCs—a limitation that can be overcome by co-culture with neonatal pericytes [27]. We therefore assessed LAMA5 deposition in OCs reconstituted with aged skin pericytes by immunostaining and hemi-desmosome/basement membrane assembly by electron microscopy in OCs seeded with neonatal keratinocytes and aged pericytes from three separate skin donors (48, 54 and 58 yo). LAMA5 deposition at the dermo-epidermal junction occurred at a much lower level in aged pericytes OCs (Figure 4A) compared to neonatal pericyte OCs (Figure 4B). Similarly co-cultures with neonatal fibroblasts did not have an inductive effect on LAMA5 deposition [27], and thus, unsurprisingly, no changes were observed when aged fibroblasts were substituted in the dermal equivalent (Figure 4B). Notably, native human skin from older donors exhibits decreased LAMA5 immunostaining (Figure 4C), which may be attributed to the observed decline in pericyte numbers in ageing skin (Figure 1) [33]. Immunostaining for other markers of the basement membrane zone, i.e., α6 integrin and collagen IV, revealed no significant differences in α6 integrin expression in OCs populated with neonatal vs. aged pericytes (data not shown); but interestingly, while both neonatal pericytes and neonatal fibroblast populated OCs exhibited strong collagen IV expression (Appendix A), it was not detectable in OCs populated with aged pericytes (48 yo, 54 yo and 58 yo) as shown in Appendix A. Notably, despite its detection in the basement membrane region in neonatal human skin (and in the dermis Appendix A), none of the OCs, even those with neonatal pericytes, exhibited deposition in the epidermal–dermal junction. These data suggest improper assembly of collagen IV in the OC model.

Ultrastructural analysis of OCs populated with aged pericytes (*n* = 3), consistently revealed a lack of hemi-desmosome and basement membrane assembly (Figure 4D), comparable to that observed with aged fibroblasts (Figure 4E, two left panels), whereas neonatal pericytes consistently conferred both features (Figure 4E) as shown previously [27].

## 4. Discussion

Our previous findings implicated dermal pericytes as potent microenvironmental regulators of neonatal human skin regeneration [26,27]. Given that keratinocytes from aged human skin exhibit poor regenerative ability both in vitro and in vivo [15,34,35], we asked whether this may be partially due to degenerative changes in the dermal microenvironment with ageing, both qualitatively or quantitatively. Our analysis of CD45^+^ versus CD45^−^ cells in aged versus neonatal skin (Figure 1A,B) was consistent with previous studies reporting an increase in the CD45^+^ dermal population using quantitative in situ immunohistological staining [36,37]—leading to the conclusion that an elevated immune response exists in the dermis with ageing. Moreover, our FACS analysis with CD45^−^VLA-1^bri^ and in situ PDGFBR immunostaining data provided strong evidence for a decline in pericyte numbers in the skin with increasing age. These data demonstrate that the cellular microenvironment of epidermal cells changes dramatically with human skin ageing with a significant drop in cellularity in the non-immune dermal cells, notably in the pro-epithelial regenerative pericyte population, particularly when comparing young versus aged skin, from the same tissue (Figure 1D,E; Appendix A). Our results confirmed a previous study reporting a decrease in dermal microvessel density, and in the pericyte to endothelial cell ratio in the ageing human skin irrespective of gender or anatomical site, based on in situ immunostaining with a different marker, i.e., a ganglioside detected by the 3G5 antibody [33]. Previous studies have also reported a decrease in the number and size of dermal fibroblasts from older donors [38] and a decline in their replicative capacity in culture [39], suggesting that this might also contribute to the lower regenerative capacity of the skin.

Functional analysis of the epidermal regenerative capacity of young versus aged pericytes in the context of neonatal fibroblasts and keratinocytes in a 3D OC reconstitution model illustrated that the introduction of 2% pericytes derived from older donors aged > 50 years old (this being the incidence in aged skin) resulted in no significant increase in epidermal regenerative capacity over fibroblasts alone (Figure 2A vs. Figure 2B) with respect to epidermal thickness; however, this could be compensated by increasing the number of pericytes to 20% an incidence found in neonatal human skin (Figure 2C–J). Although it is not possible to completely rule out the effect of difference of the anatomical site origin of the cells, these results suggest that pericytes from older skin donors exhibit a significantly decreased ability to co-operate with neonatal fibroblasts in eliciting epidermal regeneration from neonatal keratinocytes in vitro, in addition to their decreased numbers in aged skin in vivo. Ideally, we would have liked to perform OC experiments comparing young breast skin donor (20–30 yo) derived pericytes with aged donors (50 yo or above). However, it has been difficult to obtain tissue from younger subjects who tend not to undergo elective surgery as often as older ones.

Previously, we had demonstrated that in OCs, neonatal dermal pericytes were capable of stimulating more homeostatic epidermal regeneration from neonatal keratinocytes at least in part through the secretion of paracrine factors such as LAMA5 and BMP-2; specifically, pericytes restored cell polarity, planar cell divisions, LAMA5 deposition at the dermo-epidermal junction in vitro, and basement membrane/hemi-desmosome assembly not observed with fibroblasts [27]. Notably, the basal layer had a higher complement of ΔNp63^+^Ki67^+^K15^+^ in the presence of pericytes compared to fibroblasts, indicative of homeostatic maintenance of the proliferative compartment/basal layer [27]. Analysis of some of these parameters in OCs populated with pericytes from older donors in this study revealed that although some aspects of epidermal homeostasis are evident, e.g., polarised appearance of the basal layer; %Ki67+ and %ΔNp63+ basal cells (Figure 3), other features of the regenerated epidermis were absent including LAMA5 and collagen IV deposition and basement membrane/hemi-desmosome assembly (Figure 4; Appendix A).

Notably, ΔNp63 regulates both epidermal cell proliferation and differentiation-related stratification [40,41]. Interestingly, fewer suprabasal cells were ΔNp63^+^ in OCs populated with aged versus neonatal pericytes (Figure 3D), consistent with the observation that p63 null mice display fewer perpendicular/asymmetric epidermal cell divisions [42]. Thus, we hypothesise that the ability of aged pericytes to retain epidermal cells in the undifferentiated basal layer may be compromised by decreased ΔNp63 expression. Moreover, given that neonatal keratinocytes were used in these experiments, the lack of hemidesmosome/basement membrane assembly can be attributed solely to changes in the aged pericytes—especially in the context of our previous demonstration that neonatal pericyte–neonatal keratinocyte OCs display both these ultrastructural features consistently [27]. These data indicate that LAMA5 synthesis and/or deposition at the dermo-epidermal junction—a critical regulator of skin regeneration—occurs as a co-operative interaction between keratinocytes and pericytes and that its observed decrease in the ageing skin may contribute to its declined regenerative ability.

## 5. Conclusions

This study demonstrates a potential role for the dermal microenvironment particularly pericytes in the decreased skin regenerative ability observed in the human ageing skin correlated with poorer healing of skin wounds with age. It is likely that the latter is a complex multifactorial process involving various cell types found in the skin, both epidermal and dermal such as stem cell ageing, oxidative stress, and poor health due to systemic ageing not originating in the skin.

## Figures and Tables

**Figure 1 cells-10-03051-f001:**
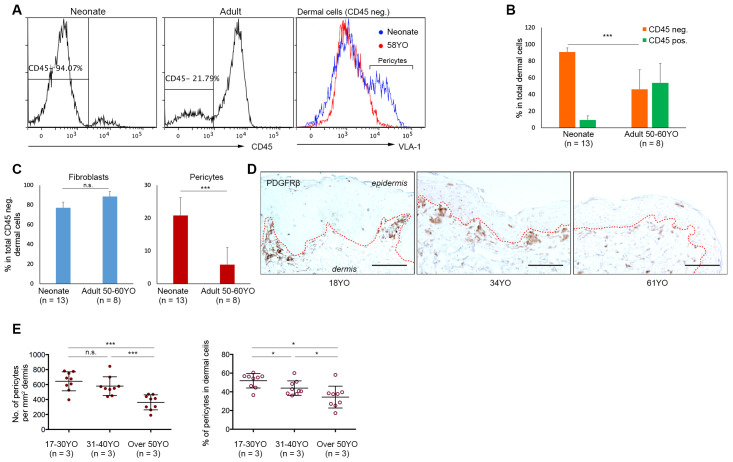
Dermal pericyte numbers decrease with increasing age in the skin. (**A**). Incidence of CD45^+^ cells is lower in neonatal (left panel) versus adult (middle panel) skin dermal cell suspensions in representative plots. Significantly lower incidence of CD45^−^CD49a^bri^ dermal pericytes in 58 yo versus neonatal skin dermal cell isolates (right panel). (**B**). Percentage of CD45^+^ versus CD45^−^ dermal cells in neonatal (*n* = 13) and aged adult skin donors aged ~50 years or over (*n* = 8 from 48–60 yo; mean of 54.6 yo) illustrating fewer CD45^−^ dermal cells in aged skin (*p* < 0.001). (**C**). Percentage of CD45^−^CD49a^dim^ fibroblasts was similar in neonatal versus adult skin (left panel; *p* = n.s.); however, a significantly lower number of CD45^−^CD49a^bri^ pericytes was evident in ageing adult skin (50–60 yo, *n* = 8) compared to neonatal skin (right panel; *p* < 0.0001). (**D**). Representative PDGFRβ immunostaining in adult breast skin from 18, 34, and 61 yo donors showing fewer PDGFRβ^+^ pericytes in aged (61 yo) skin. Scale Bar: 100μm. (**E**). Quantitative analysis of pericytes in adult breast skin from individuals aged 17–30 yo, 31–40 yo, and >50 yo expressed as number of PDGFRβ^+^ pericytes/mm^2^ dermis (left) or as %PDGFRβ^+^ of total dermal cells (right) demonstrating a decline in pericyte numbers particularly notable in skin from donors > 50 yo. Data from 9 random fields from each of 3 donors per age group. *** = *p* < 0.001, * = *p* < 0.05, n.s. = not significant.

**Figure 2 cells-10-03051-f002:**
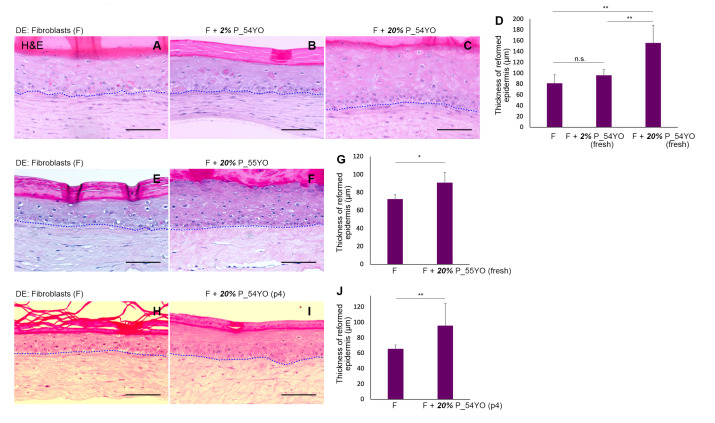
Epidermal regeneration ability of neonatal keratinocytes can be enhanced by increasing the incidence of aged pericytes in OCs. (**A**–**J**). Thickness of the epidermis reformed by neonatal keratinocytes was consistently greater when 20% dermal pericytes from aged donors (54 and 55) were included in the dermal equivalent (**D**,**E**) compared to neonatal fibroblasts alone ((**A**) vs. (**C**); (**E**) vs. (**F**); (**H**) vs. (**I**); *n* = 3). 2% 54 yo dermal pericytes (**B**) did not increase epidermal thickness over fibroblasts (**B**,**D**). Cultured pericytes (p4) were used in H–I, whereas uncultured pericytes were used in (**A**–**G**). Quantitation of epidermal thickness (**D**,**G**,**J**) was obtained from 6–8 random fields/OC. Error bars represent mean ± SD. * = *p* < 0.05, ** = *p* < 0.01, n.s. = not significant. Scale Bar: 100 μm.

**Figure 3 cells-10-03051-f003:**
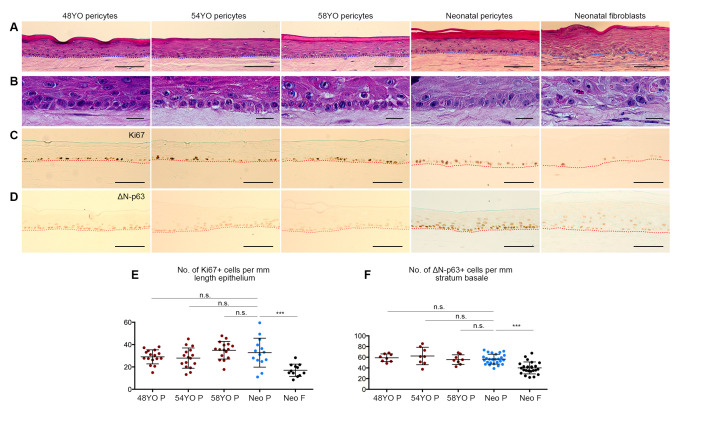
Some features of epidermal regeneration can be restored by aged pericytes in OCs. (**A**,**B**) Histological sections stained with haematoxylin and eosin; and (**C**,**D**) immunohistochemical staining of 3D skin cultures seeded with dermal pericytes from donors aged ~50–60 yo demonstrating their ability to promote epithelial tissue regeneration to a similar extent as neonatal pericytes (**A**) with re-establishment of proliferative and polarised basal keratinocytes compared to neonatal fibroblasts (**B**). Comparable Ki67 staining of basal keratinocytes was obtained with aged and neonatal pericytes but not fibroblasts (**C**); however, lower ΔN-p63^+^ staining intensity was evident in organotypics seeded with aged pericytes (**D**). Scale Bar: 100 μm. (**E**,**F**). Quantitation of Ki67 (**E**) and ΔN-p63 (**F**) immunostaining confirming no statistically significant differences (n.s.) in the numbers of Ki67^+^ and ΔN-p63^+^ basal cells obtained with neonatal versus aged pericytes, whereas their incidence was lower in fibroblast-seeded 3D cultures (*** = *p* < 0.001). Error bars represent mean ± SD. Data collected from 14–16 random fields from 2 (**E**) or 3 (**F**) independent experiments.

**Figure 4 cells-10-03051-f004:**
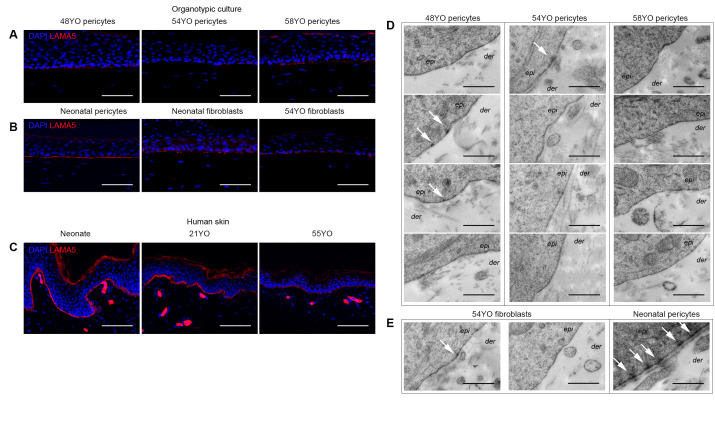
Aged pericytes do not promote LAMA5 deposition nor basement membrane/hemi-desmosome assembly at the dermal–epidermal interface in OCs. (**A**–**C**). LAMA5 immunostaining (red fluorescence; nuclei blue) in OCs seeded with aged 48–58 yo (**A**) versus neonatal pericytes; or neonatal versus aged fibroblasts (**B**); and neonatal versus 21 yo or 55 yo skin tissue in situ (**C**). (**D**,**E**). Quadruplicate transmission electron micrographs of the dermal–epidermal junctions of OCs illustrating the absence of basement membrane/hemi-desmosome assembly in OCs seeded with aged 48–58 yo skin pericytes (**D**). Similarly, aged 54 yo fibroblast co-cultures did not exhibit basement membrane assembly ((**E**), left panels) whereas neonatal pericytes did ((**E**), right panel). Epi = epidermis; der = dermis; white arrows indicate rare initiation of hemi-desmosomes in cultures with aged pericytes (**D**), fibroblasts (**D**) as compared to abundant hemi-desmosomes in cultures with neonatal pericytes ((**E**), right panel). Scale bar: 100 μm (**A**–**C**); 0.5 μm (**D**,**E**).

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
