# Peer review of "Dermal Pericytes Exhibit Declined Ability to Promote Human Skin Regeneration with Ageing in 3D Organotypic Culture Models"

_cells, 2021, doi:10.3390/cells10113051_

Round 1

Reviewer 1 Report

In this report Zhuang et al. attempted to study the effect of aged pericytes on skin organotypic cultures (OC).

The authors assessed the incidence of pericytes in human breast skin from older adult donors compared to neonatal human foreskin. They observed that pericyte numbers decrease with increasing age in the skin. This observation was confirmed by histological analysis on young and old breast skin.

Subsequently the authors showed that neonatal keratinocytes can stratified more efficiently by increasing the incidence of aged pericytes in OCs.

Comparing young pericytes and old pericytes in organotypic cultures, the authors did not observed a difference on the level of a proliferation marker as ki67 in keratinocytes. However Lama5 deposition was decreased in OCs with old compared  with young perycytes.

While the figures and the legends of this manuscript are of good quality, the conclusions related to “aging” are not supported by the data (with the exception of fig1). This is because the pericytes used in the OCs are of a different aged skin but also a different body origin. Therefore, it is not possible to distinguish between these two variables to draw the conclusions. The authors would need to back up the data with new OC experiments using pericytes from young and old breast skin samples.

In addition

-the title of Fig2 (“Epidermal tissue regeneration ability of neonatal keratinocytes can be enhanced by increasing the incidence of aged pericytes in OCs”) is an overstatement since the effect in only on keratinocytes stratification.  

- the title of Fig3 does not match the result in the figure.

Author Response

Queries raised by Rev #1:

1. While the figures and the legends of this manuscript are of good quality, the conclusions related to “aging” are not supported by the data (with the exception of fig1). This is because the pericytes used in the OCs are of a different aged skin but also a different body origin. Therefore, it is not possible to distinguish between these two variables to draw the conclusions. The authors would need to back up the data with new OC experiments using pericytes from young and old breast skin samples.

Response: We agree that ideally OC experiments should be performed with human skin from young vs aged breast skin donors. In reality, this is reliant on the availability of skin from human donors undergoing elective surgery. Unlike experiments with murine skin where the researcher has a lot of control in harvesting skin at specific ages, we are limited by what is available clinically and that we are able to access the FACS machine on that day to fractionate pericytes from a dermal isolate. Over several years of collecting breast skin tissue it has been our experience that breast skin from young donors (aged 20-30YO) is difficult to obtain. Even if we collected tissue over a couple of years, we may not be successful in sourcing tissue from younger aged donors. In contrast, neonatal foreskin from several donors (8-10) is easily available on a weekly basis in abundance, hence its use as a comparison. We have gone to some trouble to demonstrate that the trend for loss of pericyte numbers is similar in skin tissue with age irrespective of anatomical site. Moreover, supporting data showing similar changes in pericyte incidence has been reported by other labs comparing any anatomical skin site (see Helmbold et al., 2006 ref 33 in manuscript). These combined data suggest consistent changes in skin with ageing irrespective of anatomical site. We also submit that we are comparing like with like stratified epithelia sharing a similar differentiation program and not for instance mucosa with skin. We would further argue that the two other expert reviewers (Rev 2 & 3) have found our data persuasive and supportive of our hypothesis.

We have added a comment conceding that ideally experiments should be done with younger breast skin tissue and explaining the limitations of tissue availability. Page 9 of Discussion.

2. the title of Fig2 (“Epidermal tissue regeneration ability of neonatal keratinocytes can be enhanced by increasing the incidence of aged pericytes in OCs”) is an overstatement since the effect in only on keratinocytes stratification.  

Response: We have used the term “epidermal tissue regeneration” to mean keratinocyte regeneration but have now edited the title of Figure 2 to delete the word “tissue” in line with this Reviewer’s comment.

3. the title of Fig3 does not match the result in the figure.

We are not really clear as to why the reviewer thinks the title does not match the data shown. However, we have revised the title to remove the word "homeostatic" assuming that might be the term that raised concern

Reviewer 2 Report

In this manuscript Zhuang et. al. have nicely described the role of dermal pericytes on skin ageing. In addition, they have identified underlying mechanisms contributing to disturbed skin homeostasis during ageing. I would highly recommend this manuscript for publication, though I have few suggestions to improve the quality of the manuscript.

  1. Authors have shown decline in PDGFRβ positive dermal cells in aged human skin (Figure 1D), it would be interesting to analyze one or two additional dermal progenitor makers in aged human skin to find whether it is specific to pericytes or not.
  2. Authors should consistently compare the impact of neonatal, young (18y) and old (60y) pericytes/fibroblasts on epidermal homeostasis in their investigations. Also keep the group arrangement/labelling same in Figure 2 and 3, this will help the readers to follow the story.
  3. I wound suggest to analyze few additional basement components such as alpha6 integrin and collagen-IV in figure 4 to support authors conclusion.

Author Response

Reviewer #2 comments/suggestions: We thank the reviewer for their positive evaluation of our manuscript and their suggestions for improvement.

I would highly recommend this manuscript for publication, though I have few suggestions to improve the quality of the manuscript.

  1. Authors have shown decline in PDGFRβ positive dermal cells in aged human skin (Figure 1D), it would be interesting to analyze one or two additional dermal progenitor makers in aged human skin to find whether it is specific to pericytes or not.

Response: There is a paucity of dermal progenitor markers available that distinguish between them supported by our previous work Paquet-Fifield et al., 2006 J Clin Invest. Moreover, there is a lot of evidence in the literature for changes in fibroblasts with ageing in skin (see cited ref 38 Kurban et al., 1990) and in various molecular regulators such as ECM molecules – their source remains uncharacterised since good markers that distinguish between the various dermal cell types particularly human skin do not yet exist.

2. Authors should consistently compare the impact of neonatal, young (18y) and old (60y) pericytes/fibroblasts on epidermal homeostasis in their investigations. Also keep the group arrangement/labelling same in Figure 2 and 3, this will help the readers to follow the story.

Response: As explained in response to Rev#1, it is difficult to obtain human skin tissue from specific age donors and we are at the mercy of randomly aged skin donors who undergo elective surgery involving skin removal and consent to its use in research. Our preference in using human pericytes without extensive prior culture which might subject them to alterations accumulated to in vitro is a further limitation in setting up experiments with pericytes of a specific age. Unlike experiments with murine skin where the researcher has a lot of control in harvesting skin at specific ages, we are limited by wshat is available clinically.

The labels in Fig 2 are abbreviated because of the complexity of the dermal equivalent content - if we were to spell them out in full the labels would be too long to look good.  

3. I wound suggest to analyze few additional basement components such as alpha6 integrin and collagen-IV in figure 4 to support authors conclusion.

Response: We thank the Reviewer for this excellent suggestion and have analysed Collagen IV and alpha 6 integrin in the OCs generated from different aged pericytes and between neonatal fibroblast vs neonatal pericytes in response to this request. We  could not find any significant changes to a6 integrin expression associated with ageing and report it as data not shown. However, collagen IV staining showed a dramatic decline in its expression in OCs reconstituted with aged pericytes as compared to neonatal pericytes further strengthening our conclusions of the decline in homeostatic basement membrane assembly with ageing. We include this new data as Supplementary Figure 2.

Reviewer 3 Report

Zhuang and Kaur studied the ability of dermal pericytes in promoting human skin regeneration with aging in 3D organotypic culture models.  Overall all study is interesting and proved their hypothesis well with experiments.

Minor comments.

  1. Overall increase the font size in figures
  2. Figure legends contain results rather than describing what it is.

Author Response

Comments and Questions of Reviewer #3: We thank the reviewer for their positive review of our paper.

  1. Overall increase the font size in figures

Response: The size of the font is constrained by the formatting of the figures demanded by their insertion in the page size and template provided by the journal. For example, the font size is Palatino 10 as selected by the template, whereas we have used Times Roman 12 in the original manuscript. 

2. Figure legends contain results rather than describing what it is.

Response: The figure legends have been written in line with journal guidelines starting with a statement of the results shown in the figure followed by what was done in order to interpret the results obtained.

Perhaps we could get some assistance form the journal on this issue 

Round 2

Reviewer 1 Report

I recognize that can be difficult to find the best human samples and that in particular the authors struggled to find that breast skin from young donors (aged 20-30YO). However, the issue relative to the difficulty to distinguish aging vs anatomical site factors remains. I suggest to check if it is necessary to tune down several conclusions using wording such as “these data suggest” rather than “these data indicate”.

I also think that in the discussion it would be important to add  in the 319 line before  “These results…” something like : ” Although it is not possible to completely rule out the effect of difference of the anatomical site origin of the cells, these results…..”  

Author Response

Reviewer #1 comments: We thank this Reviewer for understanding the difficulty in obtaining tissue of a specific age from human subjects, particularly young adults. 

A. I suggest to check if it is necessary to tune down several conclusions using wording such as “these data suggest” rather than “these data indicate”

Response: We have carefully reviewed our claims and conclusions and are satisfied that we did not overstate our conclusions. No further changes have been made.

B. I also think that in the discussion it would be important to add  in the 319 line before  “These results…” something like : ” Although it is not possible to completely rule out the effect of difference of the anatomical site origin of the cells, these results…..”  

Response: We have made the recommended change requested by the Reviewer - shown as tracked changes in the revised manuscript.